# Room temperature organic exciton–polariton condensate in a lattice

M. Dusel[1,5 ✉], S. Betzold[1,5], O. A. Egorov[2], S. Klembt[1], J. Ohmer[3], U. Fischer [3], S. Höfling[1,4] & C. Schneider [1 ✉]

Interacting Bosons in artificial lattices have emerged as a modern platform to explore collective manybody phenomena and exotic phases of matter as well as to enable advanced on-chip simulators. On chip, exciton–polaritons emerged as a promising system to implement and study bosonic non-linear systems in lattices, demanding cryogenic temperatures. We discuss an experiment conducted on a polaritonic lattice at ambient conditions: We utilize fluorescent proteins providing ultra-stable Frenkel excitons. Their soft nature allows for mechanically shaping them in the photonic lattice. We demonstrate controlled loading of the coherent condensate in distinct orbital lattice modes of different symmetries. Finally, we explore the self-localization of the condensate in a gap-state, driven by the interplay of effective interaction and negative effective mass in our lattice. We believe that this work establishes organic polaritons as a serious contender to the well-established GaAs platform for a wide range of applications relying on coherent Bosons in lattices.

[1] Technische Physik, Physikalisches Institut and Würzburg–Dresden Cluster of Excellence ct.qmat, Universität Würzburg, Am Hubland, Würzburg 97074, Germany. [2] Institute of Condensed Matter Theory and Solid State Optics, Abbe Center of Photonics, Friedrich-Schiller-Universität Jena, Max-Wien-Platz 1, Jena 07743, Germany. [3] Department of Biochemistry, Universität Würzburg, Am Hubland, Würzburg 97074, Germany. [4] SUPA, School of Physics and Astronomy, University of St. Andrews, St. Andrews KY16 9SS, UK. [5]These authors contributed equally: M. Dusel, S. Betzold. ✉email: Marco.Dusel@uni-wuerzburg.de; Christian.Schneider@uni-wuerzburg.de

The implementation of well-controllable lattice potentials for interacting quantum particles and Bosonic condensates[1] is an engineering task of importance towards the realization of advanced classical[2] and quantum simulators[3]. The possibilities to engineer Hamiltonians in well-defined experimental settings have furthermore served as an inspiration to explore topology in synthetic systems[4]. While ultra-cold atoms in optical lattices[5], trapped ions[6], superconducting circuits[7,8], and photonic on-chip architectures[9] are considered as the leading platforms for a controlled experimental implementation and manipulation, they intrinsically suffer from a variety of serious drawbacks: the atomic approach relies on ultra-low temperatures, purely photonic approaches suffer from very small non-linearities, and superconducting circuits compose a serious technological challenge.

A hybrid approach, seeking to merge the best of all worlds, involves the implementation of strongly interacting photons on chip, in particular in the form of exciton–polaritons. Such hybrid excitation can inherit the low-loss nature of photons, and still acquire a notable non-linearity arising from the excitonic part[10,11]. Since the initial demonstration of Bose–Einstein condensation of microcavity exciton–polaritons[12], one particular focus was set on engineering the potential landscape of polaritons, in the spirit of on-chip quantum simulation[13]. While cavity polaritons and their condensates have been observed in a variety of systems, including II–VI[12], III–V semiconductors[14] and layered materials[15,16], as well as organic materials[17–19], due to the technological challenges required to precisely control their energy landscape, a vast majority of approaches toward periodic arrangements were conducted on the mature GaAs platform[13]. The degree of energy, position, coupling, and phase control has now reached such a level that it enabled the first implementation of ultra-fast simulators of the X–Y Hamiltonian[20], as well as synthetic topological Chern insulators[21] and topological lasers[22]. Even electrical injection has recently been accomplished[23]. However, these experiments still require cooling by liquid helium, and (in most cases) highly advanced nano-technology for chip processing.

In this work, we address these two fundamental complications: we utilize fluorescent proteins as active material, hosting ultra-stable Frenkel excitons. Furthermore, we directly take advantage of the soft nature of the material, and mechanically shape it into the photonic lattice environment, which has the form of a one-dimensional lattice of tightly coupled photonic hemispheric cavities. We make the following striking observations: the high quality of our device allows us to generate a close-to-ideal bandstructure of room-temperature exciton–polaritons, dictated by the photonic lattice in the tight-binding regime. Bosonic condensation is facilitated at elevated pump densities, and by shaping the pump spot, we can load the condensate into distinct lattice modes. The subtle interplay between the repulsive polaritonic interaction and localized gain provided by the pump yields the formation of a localized gap state, which we clearly identify in the real-space expansion of the condensate. We further develop a broad understanding of these nonequilibrium effects in the framework of a numerical model based on solving the Gross–Pitaevskii equation in the presence of gain, loss, and noise.

## Results

Our studied device is composed of two distributed Bragg reflectors (DBRs), which sandwich an optical spacer layer filled with a thin film of the fluorescent protein "mCherry." Our sample is fabricated as follows: first, we prepare a plateau area on the glass substrate with a depth and diameter of ~500 and 4000 μm,

respectively, by wet chemical etching. Next, lens-shaped indentations were defined by using ion beam milling (gallium ions). These hemispheric dimples have diameters ranging from 3 to 5 μm and depths between 100 and 350 nm. The shape of the micro-lenses confines the optical field to a spatial area with an effective radii of 1.0–1.5 μm, depending on the radius of curvature of the lens. Subsequently, the DBRs are deposited, which are composed of 10.5 alternating pairs of $SiO_2$ and $TiO_2$ layers with a reflectivity of 99.9% between 1.90 eV (653 nm) and 2.13 eV (582 nm). The identical layer sequence is furthermore prepared on a planar glass substrate. A layer of mCherry is coated on the DBR containing the hemispheric micro-lenses, and the planar back-mirror mechanically presses the proteins into the indentations, prior to thermally evaporating the solvents. An artistic sketch of the device is shown in Fig. 1a–c. This configuration yields calculated mode quality factors exceeding $Q_{th} = 2.0 \times 10^4$ theoretically ($Q_{exp} = 7.0 \times 10^3$ experimentally) and supports strong coupling conditions with a Rabi splitting of 240 meV, as we analyze in Supplementary Note 1. Consequently, the dispersion relation of cavity photons, and likewise, exciton–polaritons, in such a trap acquires the canonical dispersion relation of a massive particle in a harmonic trap (see Fig. 1d), featuring a ladder of dispersion-less, discrete resonances with an approximately equal energy spacing. Recently, we have discussed that the trapping of exciton–polaritons in such a structure promotes the formation of room-temperature polariton condensation with high coherence[24]. However, the controllable confinement of polaritonic quasi-particles in a single trap also composes a key building block towards the design of more advanced potential landscapes. In the presence of a neighboring hemispheric trap (see Fig. 1e), the optical modes transform into the frequency-split molecular resonances, with a well-controllable molecular coupling[25,26]. Next, we study a one-dimensional periodic arrangement of potential traps under non-resonant excitation (532 nm) in the low-density limit.

Figure 1f shows the far-field emission spectra of such a lattice (trap size 5 μm, next-neighbor distance 2.5 μm). The observed emission below the condensation threshold strongly differs from an isolated hemispheric lens, where the trapped modes are discrete in energy and momentum[24]. Evanescent coupling of the modes confined in the polariton traps leads to the formation of a distinct bandgap spectrum due to the spatial periodicity of the structure. Up to two Bloch bands are visible in the spectrum of the lower polariton, as shown in Fig. 1f. The width and curvature of the bands critically depends on the next-neighbor coupling, which is detailed in Supplementary Note 3. Resulting from the deep polariton confinement provided by the hemispheric traps, a gap between the polariton bands formed from the hybridization of s- and p-type modes evolves, which is as large as 5 meV. The experimentally determined bandgap structure of the spectrum can be fitted by a spectrum of single-particle eigenstates in a periodic one-dimensional lattice following the methodology introduced in ref. 27. Details on the model can be found in the "Methods" section, and also according to the parameters listed in Supplementary Note 2.

The high stability of the Frenkel excitons makes the non-linear regime of bosonic condensation accessible at room temperature. In the following experiment, we excite the one-dimensional lattice with a Gaussian spot (diameter ≈ 3 μm), focused on a hemispheric trap in the lattice. By increasing the pump power from $P = 0.95$ nJ/pulse over $P = 1.5$ nJ/pulse (Fig. 2b), $P = 6.0$ nJ/pulse (Fig. 2c) to $P = 29.9$ nJ/pulse (Fig. 2d), we experienced a significant qualitative modification of the lattice spectrum. First, one can capture a significant, non-linear increase in the emission intensity, centered around the spectral region on top of the Bloch

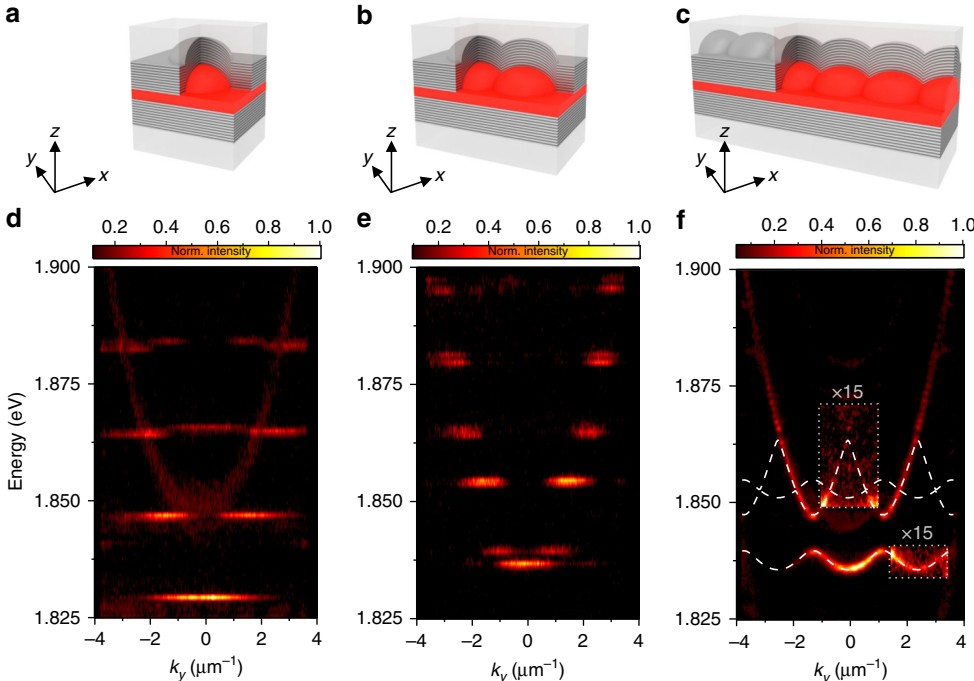

**Fig. 1 Schematic images and angle-resolved measurements of our device. a–c** Artistic illustration of single trap (**a**), molecular configuration (**b**), and one-dimensional lattice (**c**). The hemispheric indentations are filled with the "mCherry" proteins (red). **d–f** Angle-resolved photoluminescence spectrum of a single trapping site (**d**), as well as a molecular configuration (**e**) and the one-dimensional lattice (**f**), proving the formation of a bandgap polariton spectrum resulting from evanescent coupling between the sites. White lines show calculated single-particle energy bands in the effective periodic potential of the depth −270 meV and in the gray lined area the intensity is enhanced 15-fold. The measurements are recorded at a detuning between the cavity photon and exciton energy of $\Delta = E_{C} - E_{X} = -100$ meV (at $k = 0$ of the ground Bloch band). Note that slight variations in the energies of the emission spectra in **d–f** arise from a modest variation in the detuning between the structures.

band formed from photonic s-type wavefunctions. The non-linear threshold, characterized by the typical s-shape of the extracted peak area versus pump energy (see Fig. 2e, and the representation in linear scale in the Supplementary Note 4 of the manuscript) and the strong drop in the linewidth down to the resolution limit of our spectrometer, as well as the persisting blueshift of the mode above threshold, puts our system well in the framework of room-temperature bosonic condensation.

Once the condensation threshold is reached at $P = 1.5$ nJ/pulse (see Fig. 2b), in qualitative agreement with experiments conducted at 10 K in significantly more shallow potentials[28], as well as in the tight-trapping limit[29,30], the polariton condensate forms at the vicinity of the BZ edges $k_{||} = \pm\pi/a$. As the pump power is further increased, it shifts into the gap of the lattice bandstructure (Fig. 2c), and finally reaches an approximate energy offset of ≈0.8 meV with respect to the maximum of the ground band (Fig. 2d, f).

Polaritonic condensation in our lattice is not restricted to the anti-binding orbital s-mode of our lattice. It is well controllable by tuning the overlap between the pump laser and the optical mode, making our system extraordinarily versatile: as shown in Fig. 3a, similar to the scenario discussed above, the on-site pumping condition (sketched in the inset of 3a, as well as the microscopy image in Fig. 3b) loads the polaritonic condensate into the anti-binding s-mode, followed by a subsequent energy shift into the gap. The spatial coherence of this mode can be analyzed by measuring its correlation function $g^{1}(r, r')$. We use a Michelson interferometer with a variable path length in the mirror-retroreflector configuration, which overlaps the real-space luminescence from the device with its mirror image on a beam splitter, and combines them on a high-resolution CCD camera[12] (details on the technique can be found in the Supplementary Notes 8 and

9). The resulting image (Fig. 3c) shows a distinct interference pattern, in particular around the area of the pump spot. The rings in Fig. 3c provide a guide to the eye for the projection of the image and the mirror image, which have a slight vertical offset. The spatially resolved visibility of the interference fringes, plotted in Fig. 3c, reflects that the coherent mode expands over various lattice sites. The condensate phase is plotted in Fig. 3e. Since the condensate occurs at the boundary of the Brillouin zone, the phase acquires a Pi shift between neighboring lattice sites, which is in ideal agreement with the theoretical prediction outlined in Supplementary Note 5. Here, we point out that expansion of the condensate is not restricted by disorder in the sample, but, as we will explain below, by the intrinsic phenomena of gap localization. The condensation behavior can be actively controlled by shifting the position of the driving laser: off-site pumping conditions result in a significantly different phenomenological behavior at any moderate pump power above the threshold of condensation. In this setting, the condensate occurs in the binding ground state of the lattice (see Fig. 3f). We note that a similar phenomenon, yielding a transition from an anti-binding to a binding s-wave condensate, has been discussed in refs. [30,31], precisely in the framework of modified pump gain overlap for a case of a GaAs sample operated at cryogenic conditions.

Again, laterally displacing the pump laser beam, yielding an improved overlap with the p-type orbital mode of the second dispersive band, the condensation process can even be triggered on top of the bandgap (see Fig. 3g). Under these conditions, the condensate can freely expand in our lattice, and we observe a significant spatial coherence of the condensed mode over 20 µm (Supplementary Note 7).

The observed behavior of the condensed state in Figs. 2, 3b (on-site pumping) hints at the presence of a spatially localized polariton

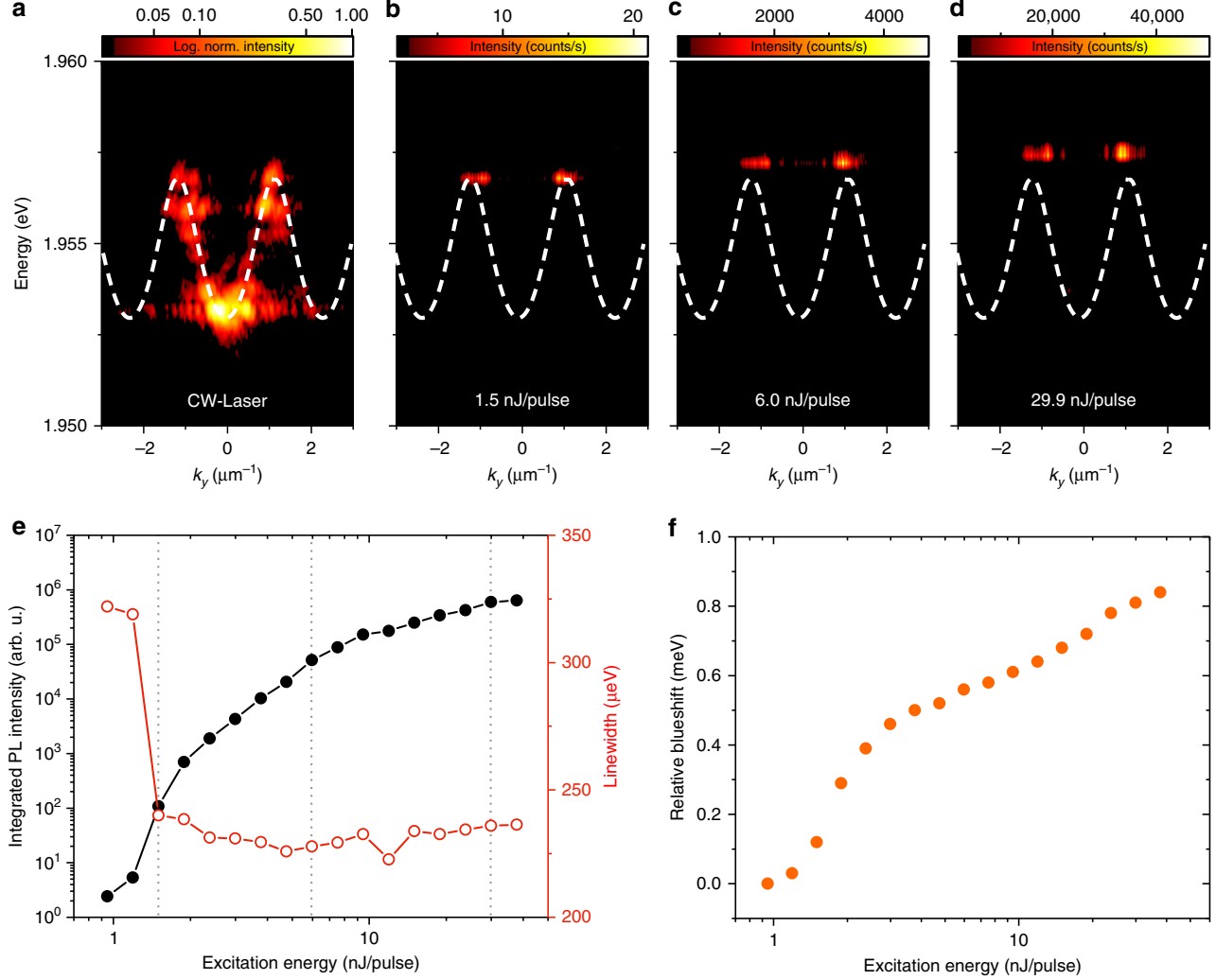

**Fig. 2 Excitation power-dependent analysis of an one-dimensional lattice. a–d** Far-field photoluminescence spectra recorded at various pump powers. Excitation with a continuous-wave laser (**a**), the white line shows calculated single-particle energy s-band in the effective periodic potential. Angled-resolved spectra for pump powers at the condensation threshold (**b** $P = 1.5$ nJ/pulse), above the threshold (**c** $P = 6.0$ nJ/pulse), and far above the threshold (**d** $P = 29.9$ nJ/pulse). The condensed mode moves into the gap. **e** Integrated emission intensity (black) and linewidth (red) versus excitation energy. At $P = 1.5$ nJ/pulse, the linewidth drops to the resolution limit of the spectrometer, whereas the intensity clearly features a superlinear increase. **f** Excitation power-dependent blueshift of the mode.

gap state, which has previously been reported in refs. [29–31] for GaAs-based exciton–polariton condensates in photonic waveguide arrays[32] and atomic Bose–Einstein condensates in optical lattices[33–35].

To further investigate this behavior, we studied the localization and expansion of the condensate in our lattice in the high-density regime. The real-space distribution of the luminescence from our device under on-site pumping conditions was projected onto a high-resolution camera. The leakage from the microcavity yields a stochastic decay of the quasi-particles, and results in a characteristic decrease of the luminescence intensity away from the pump spot. Figure 4a, b depicts the energy-resolved real-space distribution of the emitted photoluminescence. One can clearly identify the luminescence maxima from the single sites in the lattice (the directly corresponding angle-resolved measurements are depicted in Supplementary Note 6). While the pump power in Fig. 4a was chosen slightly above the onset of polaritonic condensation ($P = 2.4$ nJ/pulse), in Fig. 4b the condensation threshold was exceeded by approximately one order of magnitude ($P = 18.9$ nJ/pulse). The most dramatic effect of the condensation into the gap state can be seen from the intensity traces in Fig. 4c,

d. As the excitation power increases by one order of magnitude, the spatial extension of the condensate becomes substantially narrower. From the experimental data we extract a reduction of the spatial extension from $7.0 \pm 0.8$ μm at $P = 2.4$ nJ/pulse to $3.6 \pm 0.5$ μm at $P = 18.9$ nJ/pulse. This effect of self-localization, leading to an effective reduction of the condensate expansion, is canonical for localized gap states, which evolve in condensates shifting into forbidden bandgap, and is in an excellent agreement with our numeric model (see Supplementary Note 5).

In contrast to this behavior, off-site pumping conditions, which trigger the condensate into the lowest energy state of the s-band yield the opposite trend, namely, an increased expansion of the condensate with increasing pump (see Supplementary Note 7).

A remarkable feature in our system, which was pointed out in an earlier work[24], is the nature of the interaction in the fluorescent protein system, which is crucial for the observation of the localized gap mode[36]. While inter-particle repulsion in GaAs-based cavities is mostly induced by polariton–polariton interaction and interaction with the exciton reservoir, the density-dependent blueshift in Frenkel excitons mostly arises from the screening of the Rabi splitting with elevated density[24]. As our

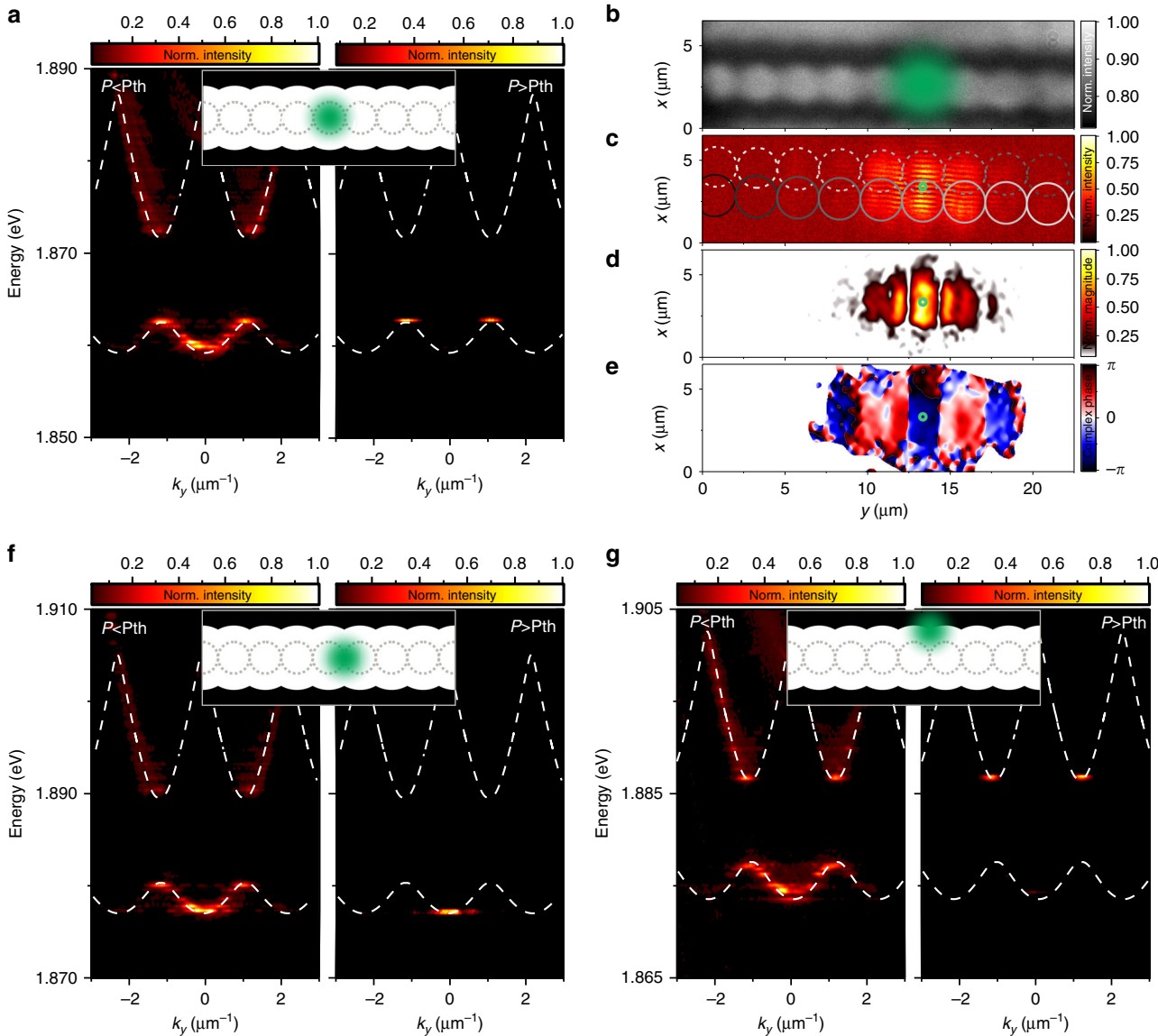

**Fig. 3 Controlled loading of the polariton condensate in the lattice. Off-site versus on-site pumping and formation of corresponding gap-state condensates. a**, **f**, **g** Far-field spectra below (left) and above (right) the condensation threshold, corresponding with the pump-potential alignment inserted with a sketch of the position of the pump laser spot (green) aligned with respect to the polariton potential. White lines show calculated single-particle energy bands. **b** Microscope image of the investigated device, indicating the positioning of the pump spot for on-site pumping. **c** Spatial coherence measurement under on-site excitation above threshold. **d** Extracted coherent magnitude and **e** phase of the condensate in the lattice. The centro-symmetric mirror point is indicated with a green circle in **c–e**.

experiments confirm, the nature of interaction is not of relevance for the observed phenomenological behavior for our system in the non-linear regime.

## Discussion

We have established the organic platform of exciton–polaritons as a new, attractive player to study bosons, and in particular non-linear bosonic condensates in optical lattices. Our platform is low cost, flexible, and can be operated at ambient conditions. Here, we already observe the full bandstructure of polaritonic Bloch bands emerging in the regime of tight binding, the controlled loading of condensates into Bloch bands with specific orbital symmetry, and the excitation of localized gap states driven by the intrinsic non-linearity of our system. Our work can be extended straightforwardly to arbitrary two-dimensional lattice geometries, as well as

synthetic time dimensions, for on-chip bosonic annealing, quantum simulation, and topological polaritonics.

## Methods

**Experimental setup**. Our optical setup can be used concurrent in near-field (spatially) and far-field (momentum space) resolved spectroscopy and imaging. We used two lasers to excite our sample. First a 532-nm continuous-wave diode laser and second a wavelength-tunable optical parametric oscillator system with ns pulses tuned to 532 nm that is resonant with the first Bragg minimum of the top mirror of our sample. For angle-resolved measurements, we used a Fourier imaging configuration. Here a Fourier lens collects the angle-dependent information in the back-focal plane of the microscope objective (×50, 0.42 NA). To obtain a spatially resolved signal, we used the Fourier configuration with an extra lens. The emission is filtered by a 550-nm longpass filter. Both the near field and the far field create an image in the focus plane of the spectrometer system. The slit in entrance plane is closed to exclude a narrow section of the image. The emitted signal is dispersed by an optical grating with 1200 lines/mm with a spectral resolution of ~230 μm for energies around 2 eV. For the Michelson measurements, the emitted signal is

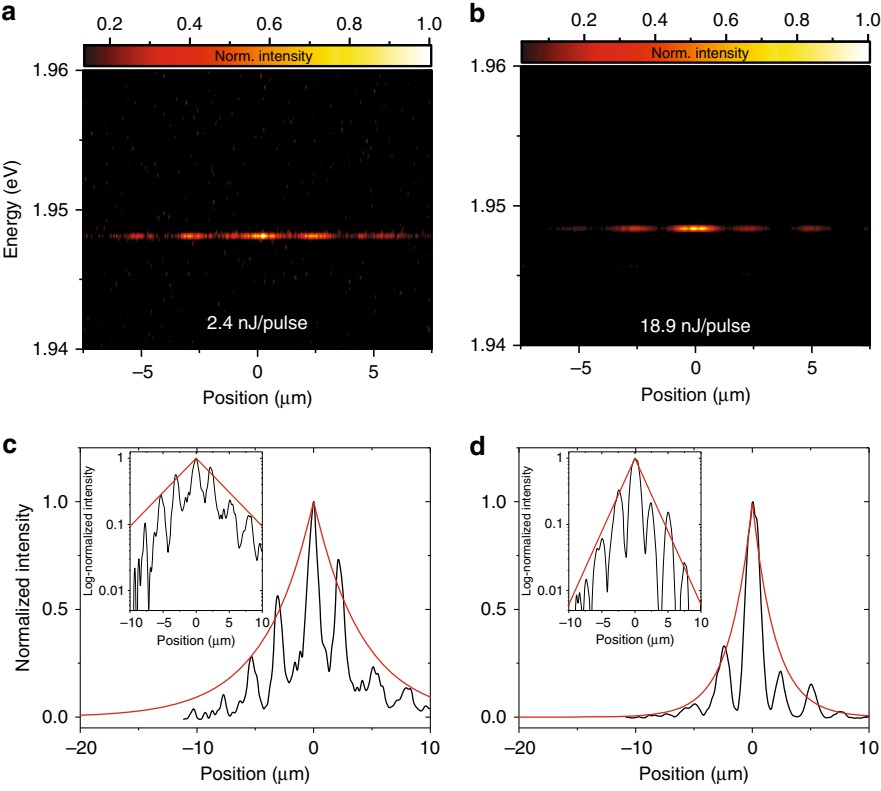

**Fig. 4 Self-localization of a gap-solitonic mode. a, b** Energy-resolved real-space images under on-site pumping conditions little above threshold (**a** $P = 2.4$ nJ/pulse) and far above threshold (**b** $P = 18.9$ nJ/pulse). **c, d** The intensity traces (black) are fitted by an exponential function (red). Narrowing of the spatial extension of the condensate from $7.0 \pm 0.8$ μm slightly above threshold to $3.6 \pm 0.5$ μm at approximately one order of magnitude above threshold characterizes the gap state. In the insets, the same profiles are shown in log scale.

deflected through a beam splitter and overlaid itself with a retroreflector and a mirror (further details can be found in Supplementary Notes 8 and 9).

**Numerical modeling.** For numerical calculations of the polariton dynamics in organic lattices, we considered a two-dimensional mean-field model, which has been widely used for simulation of inorganic semiconductor cavities[37,38], and very recently for organic systems[39]. This model consists of the open-dissipative Gross–Pitaevskii equation for the condensate wavefunction incorporating stochastic fluctuations and coupled to the rate equation for the excitonic reservoir created by the off-resonant pump[30,38].

In particular, our theoretical analysis confirms that the gap state bifurcates from the upper edge of the band characterized by the "staggered" phase, and therefore inherits the characteristic $\pi$-phase shift between neighboring density peaks[36]. For further details on numerical modeling, we refer to Supplementary Note 5.

## Data availability
The data that support the findings of this study are available from the corresponding author upon reasonable request.

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

## Acknowledgements

The Würzburg group acknowledges financial support from the state of Bavaria. Assistance by Monika Emmerling during fabrication and patterning of the sample is gratefully acknowledged. We also thank the Würzburg–Dresden Cluster of Excellence ct.qmat for financial support. S.H. also acknowledges support by the EPSRC "Hybrid Polaritonics" grant (EP/M025330/1).

## Author contributions

C.S. and S.H. initiated the study and guided the work. M.D. and S.B. designed and created the cavity devices and performed the experiments. J.O. and U.F. produced the mCherry. M.D., S.B., S.K., and C.S. analyzed the experimental data and interpreted the data supported by all coauthors. O.A.E. provided the theory. C.S., M.D., and S.B. wrote the manuscript with input from all coauthors.

## Competing interests

The authors declare no competing interests.
