## [Peer Review File · Nature Communications]

Editorial Note: This manuscript has been previously reviewed at another journal that is not operating a transparent peer review scheme. This document only contains reviewer comments and rebuttal letters for versions considered at Nature Communications .

REVIEWERS' COMMENTS:

Reviewer #2 (Remarks to the Author):

The polariton condensates in artificial lattices in semiconductor microcavities have been studied by several experimental groups before, mainly because of potential applications for optical simulators. The reliable functioning of these devices is however limited to low temperatures. The present work by Schneider and co-authors makes very important breakthrough by using organic traps. The use of proteins to hold the matter excitation permits the achievement of condensation at ambient conditions due to stability of organic excitons and the large Rabi splitting. Moreover, the one-dimensional lattice of polariton condensates demonstrate substantial band gaps, which is important for reliable control of localized solitons in the gap. The important result of this work is also in demonstration of selective excitation of the condensates.

I am satisfied with the authors reply to the last comments by the referees and the changes to the manuscript, and I recommend the paper for publication in Nature Communications in its present form.

Reviewer #4 (Remarks to the Author):

In this paper the authors present the first experiment on a polaritonic lattice at ambient conditions where fluorescent proteins play the role of the excitonic gain material (providing untr-stable Frenkel excitations).

Fluorescent proteins are used as active materials and, due to their soft nature, allows to mechanical shaping the material into the photonics lattice environment.

This platform has several interesting features cooperated to the others quantum-simulation platforms used to study the physics of interacting lattice bosons (quantum phase transitions, exotic phases and non equilibrium dynamics).

It does not require to work at cryogenic temperatures (because of the stability of organic excitons and the large Rabi splitting) and it can represent a good option to the well-established GaAs systems in terms of flexibility and cost effectiveness.

I found the paper very well written, clear and accessible to a broad audience of physics, both theoreticians and experimentalists.

I think it is important to propose novel platforms for quantum simulation of extended (open) lattice systems since each of them suffers severe limitations. Thus, to understand different physical phenomena of interacting lattice bosons, simulators with different features are needed.

Different communities can be interested in this work as researchers interested in quantum simulation, exciton-polariton condensation, open quantum systems, driven-dissipative systems, nonequilibrium lattice physics.

This manuscript has been already reviewed by different Referees. Their comments were pertinent and addressed also the technical aspect of this work. I carefully read the (second) reply to the Referees and considered the modifications made to the Manuscript.

I consider them well done and complete. In conclusion, in my opinion, the present paper is suitable for the publication in Nature Communications in its present form.